# Precursor-free eruption triggered by edifice rupture at Nyiragongo volcano

D. Smittarello[1 ✉], B. Smets[2,3], J. Barrière[1], C. Michellier[2], A. Oth[1], T. Shreve[4], R. Grandin[5], N. Theys[6], H. Brenot[6], V. Cayol[7], P. Allard[5], C. Caudron[8], O. Chevrel[7], F. Darchambeau[9], P. de Buyl[10], L. Delhaye[2,3], D. Derauw[11,12], G. Ganci[13], H. Geirsson[14], E. Kamate Kaleghetso[15,16,17], J. Kambale Makundi[18], I. Kambale Nguomoja[19], C. Kasereka Mahinda[16], M. Kervyn[3], C. Kimanuka Ruriho[20], H. Le Mével[4], S. Molendijk[15], O. Namur[15], S. Poppe[8,21], M. Schmid[22], J. Subira[1,2,16,23], C. Wauthier[24,25], M. Yalire[16], N. d'Oreye[1,26], F. Kervyn[2] & A. Syavulisembo Muhindo[16]

Classical mechanisms of volcanic eruptions mostly involve pressure buildup and magma ascent towards the surface[1]. Such processes produce geophysical and geochemical signals that may be detected and interpreted as eruption precursors[1–3]. On 22 May 2021, Mount Nyiragongo (Democratic Republic of the Congo), an open-vent volcano with a persistent lava lake perched within its summit crater, shook up this interpretation by producing an approximately six-hour-long flank eruption without apparent precursors, followed–rather than preceded–by lateral magma motion into the crust. Here we show that this reversed sequence was most likely initiated by a rupture of the edifice, producing deadly lava flows and triggering a voluminous 25-km-long dyke intrusion. The dyke propagated southwards at very shallow depth (less than 500 m) underneath the cities of Goma (Democratic Republic of the Congo) and Gisenyi (Rwanda), as well as Lake Kivu. This volcanic crisis raises new questions about the mechanisms controlling such eruptions and the possibility of facing substantially more hazardous events, such as effusions within densely urbanized areas, phreato-magmatism or a limnic eruption from the gas-rich Lake Kivu. It also more generally highlights the challenges faced with open-vent volcanoes for monitoring, early detection and risk management when a significant volume of magma is stored close to the surface.

An 'open-vent' volcano is a system in which a conduit connected to a magma reservoir allows gas and/or magma to reach the ground surface[4,5]. This usually limits pressure buildup in the plumbing system, favouring persistent effusive or strombolian eruptive activity. In such a context, detecting geophysical precursors of an impending eruption is complicated by the presence of background signals related to that persistent activity[6]. Located in the western branch of the East African Rift, in the Congolese part of the Virunga Volcanic Province, Mount Nyiragongo and its neighbour, Nyamulagira, are amongst the most active volcanoes on Earth[7]. Nyiragongo (3,470 m above sea level (a.s.l.)) is a typical open system: its summit hosts a large (200–250 m

wide), highly fluid lava lake that has been characteristic of its eruptive activity since at least 1928[8–10]. Nyiragongo represents a direct threat[11] to the nearby cities of Goma (with more than 1 million inhabitants[12]) in the Democratic Republic of the Congo (DR Congo) and Gisenyi (with around 110,000 inhabitants[12]) in Rwanda, located at 1,500 m a.s.l. (Fig. 1). Before the volcanic crisis in May 2021, the first ever monitored in real time at Nyiragongo, only two flank eruptions were historically described, in 1977[9,13] and 2002[14,15]. Both were characterized by similar pre-eruptive forerunners, that is, a nearby $M > 5$ earthquake in the rift valley, a voluminous effusive eruption of Nyamulagira and changes in the local seismic activity, with earthquakes felt in Goma and Gisenyi a

[1]European Center for Geodynamics and Seismology, Walferdange, Grand Duchy of Luxembourg. [2]Department of Earth Sciences, Royal Museum for Central Africa, Tervuren, Belgium. [3]Department of Geography, Vrije Universiteit Brussel, Brussels, Belgium. [4]Earth and Planets Laboratory, Carnegie Institution for Science, Washington, DC, USA. [5]Université Paris Cité, Institut de Physique du Globe de Paris, CNRS, Paris, France. [6]Royal Belgian Institute for Space Aeronomy (BIRA-IASB), Brussels, Belgium. [7]Université Clermont Auvergne, CNRS, IRD, OPGC, Laboratoire Magmas et Volcans, Clermont-Ferrand, France. [8]Laboratoire G-Time, Department of Geoscience, Environment and Society, Université libre de Bruxelles, Brussels, Belgium. [9]ContourGlobal/KivuWatt Ltd, Kibuye, Rwanda. [10]Royal Meteorological Institute of Belgium, Brussels, Belgium. [11]Centre Spatial de Liège, Université de Liège, Angleur, Belgium. [12]Universidad Nacional de Río Negro, Instituto de Investigación en Paleobiología y Geología de Río Negro-CONICET, General Roca, Argentina. [13]Istituto Nazionale di Geofisica e Vulcanologia, Osservatorio Etneo, Catania, Italy. [14]Institute of Earth Sciences, University of Iceland, Reykjavík, Iceland. [15]Department of Earth and Planetary Sciences, KU Leuven University, Heverlee, Belgium. [16]Goma Volcano Observatory, Goma, Democratic Republic of the Congo. [17]Département de Géologie, Université de Goma, Goma, Democratic Republic of the Congo. [18]Protection civile au Nord Kivu, Goma, Democratic Republic of the Congo. [19]Protection civile de Goma, Goma, Democratic Republic of the Congo. [20]Institut National de la Statistique Nord-Kivu, Goma, Democratic Republic of the Congo. [21]Centrum Badań Kosmicznych Polskiej Akademii Nauk (CBK PAN), Warszawa, Poland. [22]Eawag, Swiss Federal Institute of Aquatic Science and Technology, Surface Waters—Research and Management, Kastanienbaum, Switzerland. [23]Department of Geography, Université de Liège, Liège, Belgium. [24]Department of Geosciences, The Pennsylvania State University, University Park, PA, USA. [25]Institute for Computational and Data Sciences, The Pennsylvania State University, University Park, PA, USA. [26]Department of Geophysics/Astrophysics, National Museum of Natural History, Walferdange, Grand Duchy of Luxembourg. ✉e-mail: delphine.smittarello@ecgs.lu

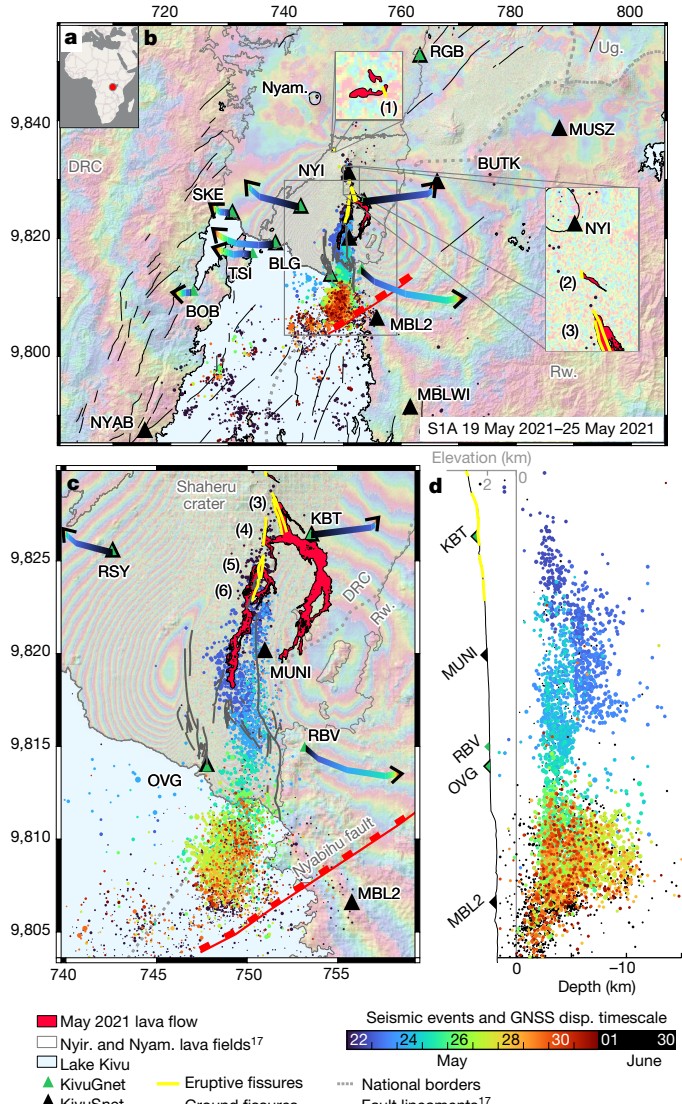

**Fig. 1 | Co-eruptive geodetic signals and seismicity. a**, Situation map. **b**,**c**, Sentinel-1 (S1) 19 May 2021 to 31 May 2021 ascending (A) interferogram overlaid with automatic earthquake locations and GNSS displacements (disp.) over time (blue to black colours with time from the onset of the eruption), eruptive fissures (yellow lines 1 to 6, from north to south), ground fissures detected from interferogram discontinuities (grey lines), lava flows (red area) and seismic and GNSS stations from KivuSNet[18] and KivuGNet[25] available during the crisis (black and green triangles, respectively). DRC, Democratic Republic of the Congo; Nyam., Nyamulagira; Rw., Rwanda; Ug., Uganda. Panel **c** shows a magnification of the central box in **b**. **d**, North–south transect of hypocentral depth (same symbols as in **b** and **c**). Coordinates are given in kilometres in the WGS 1984 UTM (Zone 35S) system.

few days before the eruption onset. Vigorous variations of the lava lake level were observed a few weeks before the 1977 eruption[13]. The lava lake had been encrusted since December 1995, and a dark plume from the summit crater and rumbling sounds accompanied ground vibrations before the 2002 eruption[16]. Those eruptions activated the same north–south eruptive fissure network, along the southern flank of the volcano (Extended Data Fig. 1), and lava flows followed similar paths[17] (Extended Data Fig. 2). Volcano monitoring and contingency planning in Goma were developed on the basis of this very limited knowledge of Nyiragongo's eruptive history, prioritizing scenarios of 1977/2002-like eruptions, with clear precursory signals detectable a few days to a few weeks before the event.

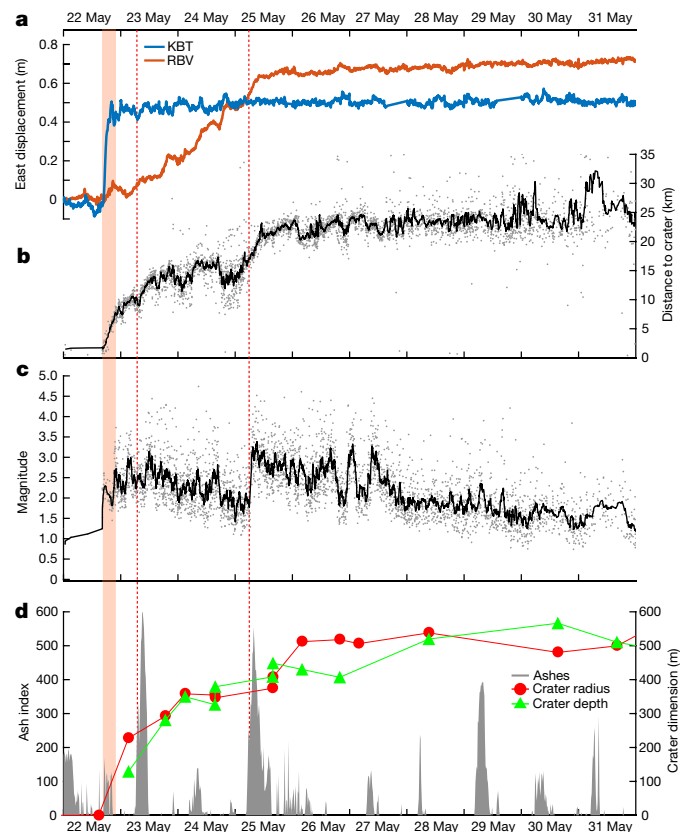

**Fig. 2 | Time series of co- and post-eruptive signals. a**, Eastward displacement recorded at KBT (blue) and RBV (orange) GNSS stations. **b**,**c**, Distance to crater (**b**) and magnitude (**c**) of automatically located earthquakes (grey dots) overlaid with a 15-event moving average (black lines). **d**, Volcanic ash index (from SEVIRI InfraRed satellite) and crater dimensions from SAR imagery analysis. Crater depth is estimated with reference to the 2002 platform at around 3,190 m (Methods). The around 6 h effusive activity and main ash column emission at the summit (23 and 25 May) are marked by the red shading and dotted lines, respectively.

Nyiragongo volcano started erupting in the early evening of 22 May 2021. A retrospective analysis of the seismic and acoustic signals recorded by the KivuSNet network[18] enables the sequence that led to the beginning of that eruption to be unravelled. Since the network installation in 2015, a clear broadband (more than 0.4 Hz) continuous seismo-acoustic tremor has been measured, tracking the persistent feeding, degassing and convection of fresh magma within the lava lake[10,19]. At around 15:57 UTC, an increasing high-frequency ground motion signal started to mask that tremor at the summit station NYI (Extended Data Fig. 3). At 16:15, the first volcano-tectonic earthquakes located at shallow depth within the edifice were detected across the local network (Methods). From 16:35, the seismic activity was accompanied by an increasing acoustic level with coherent infrasound sources attributable to Nyiragongo. Finally, between 16:46 and 16:56, bursts of continuous infrasound signals (Extended Data Fig. 3) roughly coincided with the first visual accounts of lava outflows, marking the onset of the 2021 effusive eruption on the south-eastern flank of Nyiragongo.

The main eruptive fissures opened in a north–south sequence from higher to lower altitudes, between the Shaheru Crater and Mujoga village (fissures 2–6, Fig. 1 and Extended Data Fig. 1). A shorter fissure emitting a small lava flow on Nyiragongo's north-west flank was confirmed later using airborne thermal imagery (fissure 1, Fig. 1a), but the exact timing of the onset of effusion is unknown. Field observations and petrographic analysis (Methods) show that the lava flows contained relatively few millimetre-scale bubbles, indicating a degassed magma.

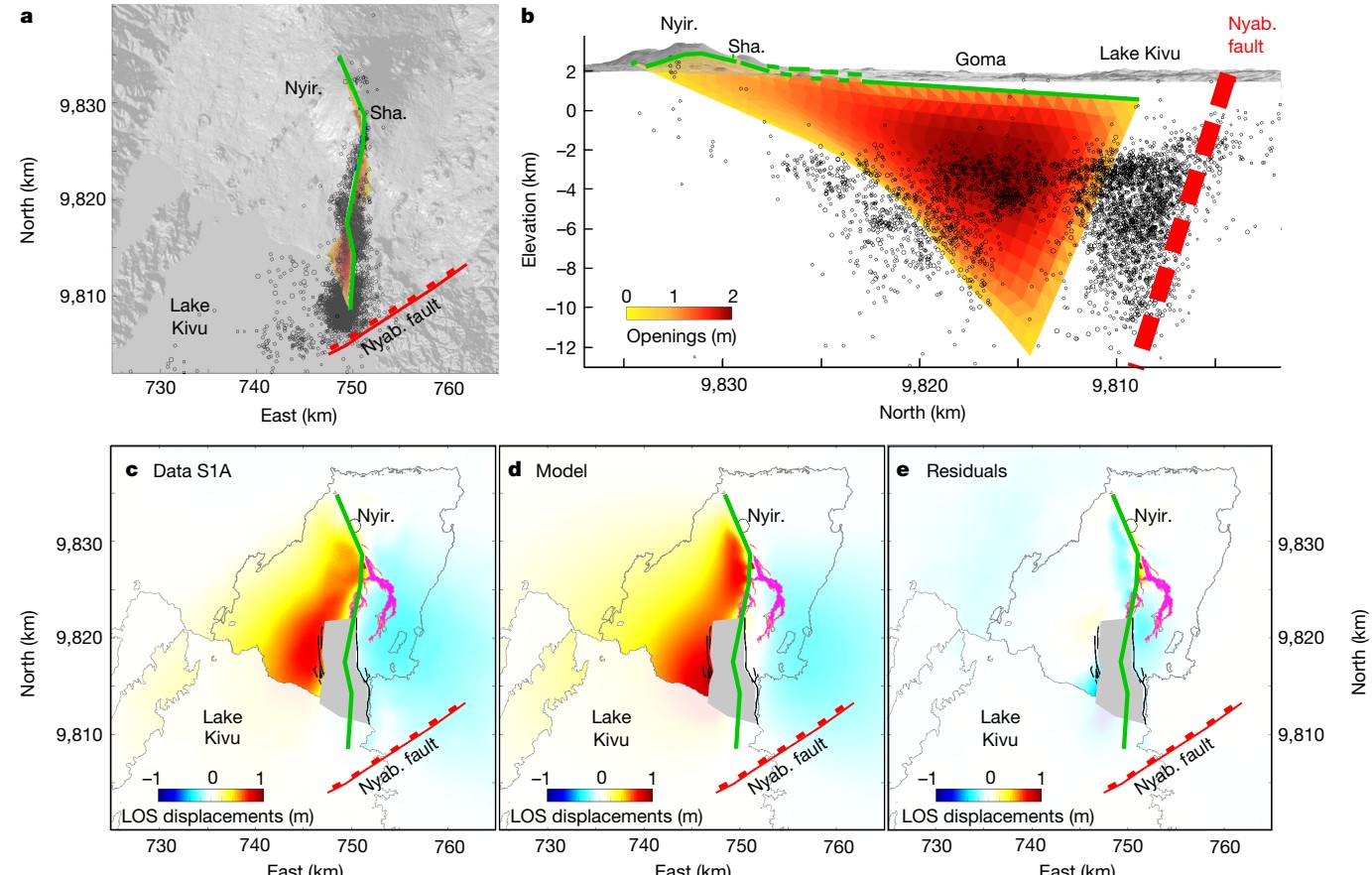

**Fig. 3 | Inversion results. a,b**, Best results of dyke geometry inverted from four interferograms spanning the eruption overlaid with seismicity between 22 and 31 May in a map view (**a**) and along a north–south cross section (**b**) (see also Extended Data Fig. 1). Colours represent the dyke opening (0–2.5 m). Sha., Shaheru Crater. **c**, Displacement map observed in satellite line of sight (LOS) from the S1 interferogram shown in Fig. 1. **d**, Modelled displacement in LOS. **e**, Residuals. In **c**–**e**, the lava flows are mapped in pink. Inelastic deformation within the graben is masked in grey. Black and green lines represent ground fissures and the dyke top trace, respectively. The dyke top trace is connected to eruptive fractures (**b**). Nyabihu Fault is marked in red. Its 72.5° dip is estimated from seismic profiles[45]. Coordinates are given in kilometres in the WGS 1984 UTM (Zone 35S) system.

The mineralogical and chemical composition of these lavas, as well as their crystal cargo, are similar to those of the 2002 lava flows and lava emitted by an intracrater summit vent that had formed in 2016 (Extended Data Fig. 4). This indicates that the 2021 lava flows were most likely fed by the shallower part of the magma plumbing system. However, some highly primitive olivine crystals were observed in the 2021 lavas, which must originate from deeper parts of the plumbing system.

Most lava was emitted from fissures 3 to 6 (Fig. 1c). Lava from fissures 3 and 4 flowed eastwards and cut the National Road between Goma and Kibati for a length of over 850 m before turning southwards, following the topography and crossing the Rwanda/DRC border. By contrast, lava from fissures 5 and 6 mainly went southwards, reaching the suburbs of Goma, and stopped 1.3 km north of the Goma International Airport runway (Extended Data Fig. 2). Lava covered around 10 km², which, assuming an average flow thickness of 1–1.5 m, corresponds to an estimated volume of 10–15 Mm³. Lava flows destroyed eight schools, three health centres, a dozen churches, several key economic infrastructure elements, and electrical and telecommunications facilities. They also damaged the largest water tank supplying the northern part of Goma, depriving nearly 550,000 people of access to water. According to a field survey conducted in the immediate aftermath of the eruption, approximately 6,000 households were left homeless (Methods). Furthermore, approximately 220 people died or were reported missing as recorded in our survey, and more than 750 people were reported to be injured.

The effusive activity lasted around 6 h, in agreement with infrasound data. However, the seismic crisis developed for about ten days. After a lull on 24 May, it culminated with a series of 271 events of $M > 3$ on 25–26 May, among which 20 were of $M > 4$ (Figs. 1 and 2). Synthetic aperture radar (SAR) interferograms spanning the whole crisis and acquired along both ascending and descending orbits show a three-lobe pattern. The western and eastern lobes correspond to an east–west opening associated with an uplift. The central lobe on Goma city depicts a subsiding area oriented north–south. Such a deformation pattern is typical of a dyke opening accompanied by graben formation[20]. In Goma and Gisenyi, newly opened fissures marked the edges of this 450-m-wide graben (Methods). Interferograms were inverted using the elastic model DEFVOLC[21,22] (Methods). The inversion results in a subvertical shallow dyke of volume about 240 Mm³ with a maximum opening of 2.5 m. It extends laterally over 4 km northwestwards from Nyiragongo Crater and over 25 km southwards (Fig. 3). The top of the dyke is estimated to be at a depth of around 450 m below the surface (Extended Data Figs. 5 and 6 and Extended Data Table 1). This agrees with the value of 390–480 m computed from a simple trigonometry assumption for a 450-m-wide graben bordered by 60–65° dipping faults[23,24]. Seismic and geodetic (global navigation satellite systems (GNSS)[25] and radar interferometry) data show the gradual southward migration of the dyke, from Nyiragongo towards Goma airport on 23–24 May, to Goma and Gisenyi city centres on 25–26 May and, finally, to the northern basin of Lake Kivu from 27 May onwards. Seismicity remained

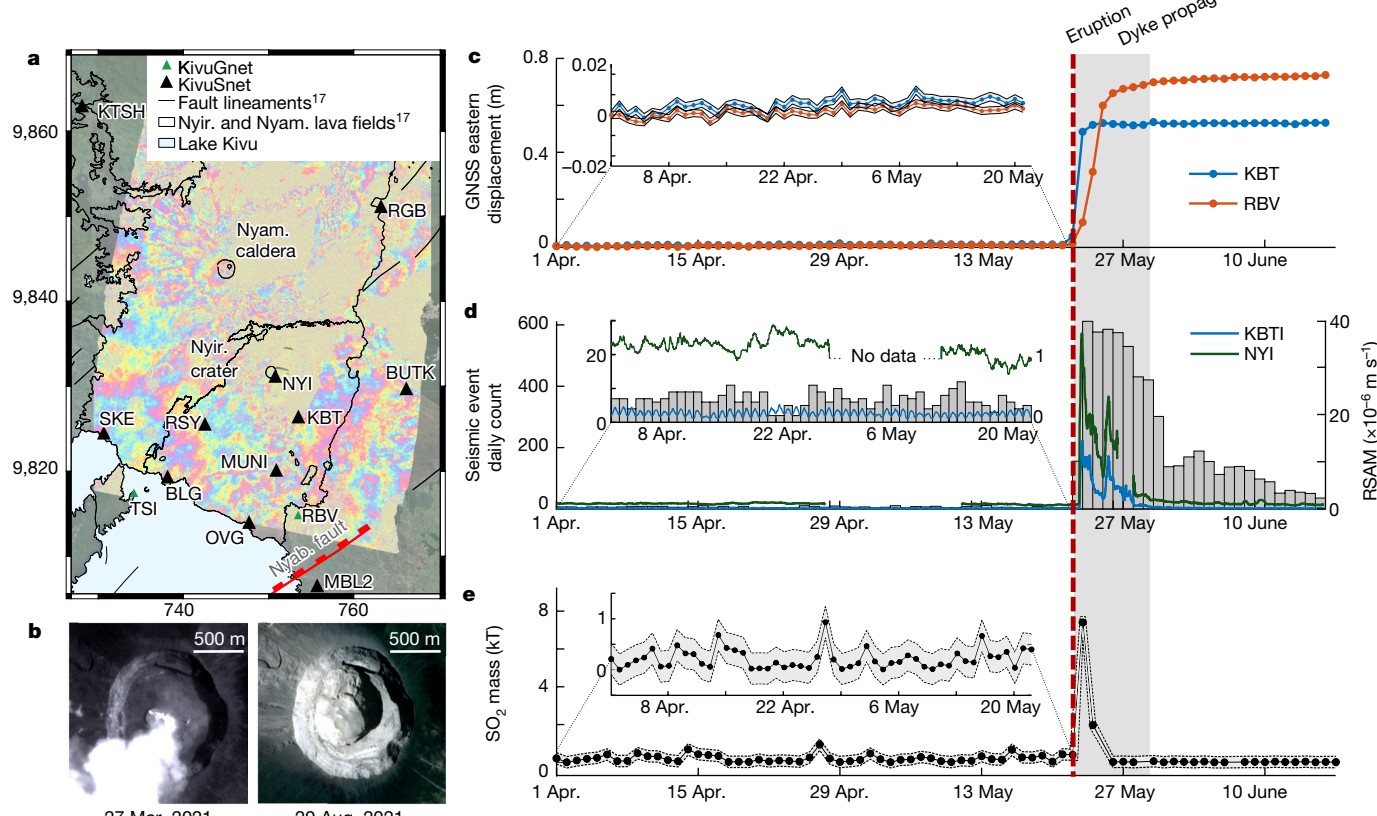

**Fig. 4 | Absence of precursory signals. a**, COSMO-SkyMed (CSK) interferogram from 21 May 2021 at 15:37 UTC to 22 May 2021 at 15:37 UTC showing no obvious deformation less than 1 h before the eruption starts. Coordinates are given in kilometres in the WGS 1984 UTM (Zone 35S) system. **b**, PlanetScope[46] image comparison of Nyiragongo Crater between 27 March 2021 and 9 August 2021. **c**, Daily eastern displacements recorded at KBT (blue) and RBV (orange) permanent GNSS stations (see location on map **a**) from 1 April 2021 to 30 June 2021. Error bounds represent 2 standard errors. **d**, Daily count of seismic events automatically detected and located (fulfilling selection criteria defined in the Methods) from 1 April 2021 to 30 June 2021 and 12-h moving median of real-time seismic amplitude measurement (RSAM) filtered between 2 Hz and 10 Hz at NYI (green) and KBTI (blue) permanent seismic stations. Note that the KBTI station is co-located with the KBT GNSS station. **e**, $SO_2$ mass automatic detection from TROPOMI over the Virunga region. Error bounds represent 2 standard errors.

intense until 31 May, with 27 additional events with $M > 3$, then progressively calmed. In addition, we tracked a gradual collapse of the crater floor over 500–550 m deep using daily radar imagery (Fig. 2), combining images from 8 satellites and 15 acquisition geometries (Methods and Extended Data Table 2). Two several-kilometre-high ash columns were emitted, at 06:00 on 23 May and at 04:30 on 25 May (Methods). The latter was associated with a sudden jump in the average magnitude of seismic events (Fig. 2). The crater walls were unstable and a widening of the crater was measured a few hours after each ash column emission (Fig. 2).

Similar to other dyke intrusions[26], seismicity appears to be localized mainly at the bottom edge of the propagating dyke as well as in the area of maximum opening below Goma (Fig. 3). According to the spatial migration of seismicity (Fig. 2), the dyke progressed in three main steps. The first step on 22 May consisted of a lateral propagation within the edifice. Starting from the lava lake system, it extended over approximately 7 km, erupting in several locations. The second step began on 23 May with the 06:00 ash emission. The seismic swarm progressed southwards and paused on 24 May. The third step began on 25 May simultaneously with the 04:30 ash emission. The progression finally stopped below Lake Kivu at the vicinity of the Nyabihu Fault on 27 May. However, the seismicity in the Kivu basin remained above the usual background[18] for several months.

Ground-based and remote-sensing monitoring techniques offer a detailed view of the whole sequence of events, challenging previous

views on the mechanisms controlling eruptions at Nyiragongo. Despite multiparametric monitoring, no precursor signal were identified during the months, weeks, days and even hours before the eruption. Seismicity was at background level[18,19] (Fig. 4), except for two seismic swarms in November 2020 below Lake Kivu and mid-April 2021 below Nyamulagira volcano (long-period events). Both swarms have no obvious link with the 2021 Nyiragongo eruption. No significant pre-eruptive deformation was detected either in GNSS time series or in radar interferograms, even with an image acquired less than an hour before the eruption started (Fig. 4). Sulphur dioxide ($SO_2$) degassing (Methods) was also at background level (Fig. 4) as was thermal activity (Methods and Extended Data Fig. 7). Lava filling of Nyiragongo Crater has been ongoing intermittently since the 2002 eruption[27,28]. Beginning in 2020, lava started to inundate the remnant platform of the 2002 crater collapse, but was still around 85 m below the level reached before the 1977 eruption. This observation was a matter of concern[27] but did not allow the forecasting of the 2021 eruption. In contrast to the the pre-2002 situation, in which the lava lake had been encrusted since 1995, the lava lake was actively convecting before the eruption, implying a steady magma and gas supply from the deep (more than 10 km) magma plumbing system[28], which is demonstrated by our finding of deeply derived olivine crystals within the lava samples.

Unlike the 1977 and 2002 eruptions, which had a tectonic trigger[13–15], we infer that the 2021 eruption was the consequence of an edifice rupture, which could have resulted either from stress reaching tensile

strength or from time-dependent weakening owing to sustained stresses and elevated temperatures. Being already close to the surface, the magma only had to propagate laterally for a short distance before erupting, leaving little time to detect and interpret the associated signals. The eruption started less than 40 min after the first detectable anomalous seismic event. Even for the best prepared volcano observatories, such a delay is extremely short, yet it would allow for some risk-reducing actions using appropriate monitoring infrastructure (for example, denser instrumentation of the edifice), early warning systems and robust emergency procedures, which were not available at Nyiragongo in May 2021.

Dyke intrusion is a common mode of transport for large volumes of magma moving laterally[29]. Dyke migration along the rift has already occurred at Nyiragongo in the recent past[15] and will occur again. Although tracking the seismic and geodetic signals produced by the progression of a dyke is possible after its onset, it remains a challenge to predict if, where and when it could erupt[29]. The 2021 eruption at Nyiragongo shares similarities with recent eruptions at other volcanoes. At Kilauea, Hawaii, signs of long-term pressure buildup were observed[30], but there was no clear precursor of the rupture that initiated the voluminous dyke intrusion into the Lower-East Rift Zone in 2018[31]. Massive drainage of the central feeding systems caused long-lived lava lake extinction in 2018 at Ambrym volcano on Vanuatu[32] and at Kilauea[31]. It also caused caldera subsidence at Bardarbunga in Iceland in 2014[33] as well as catastrophic collapse at Piton de la Fournaise in 2007[34] and Kilauea in 2018[31]. Most of those analogous dyke propagations produced protracted and voluminous eruptions at the extremity of the dyke, kilometres away from the central feeding zone[31,33], even when a first short eruptive stage occurred at a higher elevation[32,34]. Because dyke propagation happens below the cities of Goma and Gisenyi as well as below Lake Kivu, it has the potential to trigger a variety of catastrophic events: fast-running flows in a densely populated agglomeration as in 2002[14], phreato-magmatic explosions[35] or even a limnic eruption of the gas-rich Lake Kivu[36]. Nevertheless, numerous factors may influence the dyke propagation and hence its likelihood of producing an eruption. At Nyiragongo, a low buoyancy contrast is expected between highly degassed magma and the edifice rocks. Intrusion events without a subsequent eruption in Iceland and the Afar have shown that such a low buoyancy contrast may effectively prevent dykes from reaching the surface[37,38]. Tectonic extension of the Kivu rift may have further helped the sucking of magma down the rift[39], as the edifice broke open, a mechanism which has been illustrated by laboratory experiments[40]. In additional, pre-existing faults crossing the dyke trajectory may act as a stress barrier and stop or re-orient the dyke progression as was inferred in 1975–1987 at Krafla, Iceland[39], or in 2000 at Miyakejima, Japan[41]. At Nyiragongo, the regional Nyabihu Fault most likely arrested the southward propagation of the dyke below the north-eastern basin of Lake Kivu.

Among the hazards entailed, the impact of a magma intrusion within Lake Kivu is especially understudied. It was observed in 2002 that a modest lava flow entering the lake at shallow depth was not able to trigger a limnic eruption[42], but the potential impact of a sublacustrine eruption remains unknown. The 2021 dyke intruded at very shallow depth (450 m versus 3,000 m for the dyke associated with the 2002 eruption[15]), which makes the scenarios of the dyke erupting within the cities or the lake more probable than previously thought.

The May 2021 eruption pleads in favour of in-depth studies of these potential consequences before the next volcanic crisis happens at Nyiragongo. A probabilistic quantification of the associated risks is required, but remains highly challenging, in view of the limited knowledge of the eruption history of Nyiragongo. Determining the amount and speed of lava and gases that could potentially enter the lake in a future subaquatic eruption is the key to evaluating how likely it is that a dyke erupting within the lake could either (1) not create any relevant gas exsolution from the lake waters, (2) create a short-lived local gas release close to the site of the intrusion or (3) create a large-scale limnic eruption. Future work should also investigate whether the presence of magma at shallow depth may cause any changes in the dynamics of subaquatic springs feeding the lake, as these are the main drivers for its vertical density structure[36].

The real-time tracking of the shallow dyke at Nyiragongo demonstrates the feasibility of following magma movements using adequate monitoring techniques even in an environment as challenging as the Kivu region, which is recurrently affected by armed conflicts and related rural–urban population migrations[43,44]. However, the 2021 crisis highlighted the urgent need to prepare for additional hazards. Magma storage close to the surface in open systems means that eruptions may occur with only very short-term precursory activity, thus raising major challenges for their monitoring that need to be adequately addressed.

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

# Methods

## Seismicity analysis

On 22 May 2021, nine broadband telemetered seismic stations of the KivuSNet network[18] were operational in a radius of 65 km around Nyiragongo, including the stations NYI at the summit (around 3,410 m a.s.l.) and KBTI, 6 km away on the south-eastern flank (around 2,000 m a.s.l.), both also equipped with infrasound sensors. Note that KBTI station is co-located with KBT GNSS station (Fig. 1). For security reasons, most stations are deployed in protected sites (for example, United Nations military camps) close to human activity, thus leading to better network performance for seismic event detection and location during the night time. Located on the edge of Nyiragongo's crater rim, unaffected by anthropogenic diurnal noise, the summit station NYI station is the most suitable to detect the onset of seismic activity related to the eruption.

Seismic events are detected and located using a source-scanning (grid-search) method enabling robust automatic location solutions and a better evaluation of the location error than traditional iterative inversion techniques based on manual/automatic picking of P- and S-wave arrivals[47]. The main steps of this interferometric-based location procedure are (1) computing three-dimensional (3D) seismic (P- and S-wave) differential travel-time grids for each station pair using the local velocity model[48], (2) computing P- and S-waves characteristic waveforms (CWs) from raw data at each station (kurtosis or envelope, respectively), (3) calculating cross-correlation functions of P- and S-wave CWs for each station pair, (4) projecting the absolute amplitude of cross-correlation functions (that is, correlation coefficients) obtained at differential travel times corresponding to each 3D grid node for each station pair, and (5) stacking all 3D 'correlation coefficients' grids to get the maximum likelihood location (for example, 12 P-wave kurtosis waveforms obtained at 12 stations would give 66 station pairs, so 66 spatial grids to stack to get the final location likelihood). We also use cross pairs of P- and S-wave CWs in addition to single phase P or S station pairs to better constrain the final hypocentre solution[49]. It is important to note that cross-correlating waveforms between station pairs removes the need (and the possibility) to get an origin time for the seismic event. Therefore, we determine this origin time a posteriori from the best source location obtained after the cross-correlation procedure. To do this, we fit synthetic CWs corresponding to the obtained hypocentre solution with the 'observed' CWs (that is, computed from the raw waveforms). The time difference between the maximum of each observed CW and its associated 'calculated' arrival time (obtained from the synthetic CWs) enables an estimate of the origin time and a value of the root-mean-square error (in seconds) for the final solution.

In particular, the depth estimates are strongly dependent on the chosen local velocity model. Nonetheless, tests with different simplified layered models showed that shallow seismic sources related to the dyke intrusion are a robust feature. Some important quality criteria for selecting the most accurate event locations are (1) at least six stations available, (2) a minimum of seven good observations (P or S wave) and (3) an azimuthal gap of less than 180°. Good observations are selected if there is less than 1 s between the maxima of observed CW and theoretical CW determined a posteriori from the best source location on the 3D travel-time grid. An additional condition is that at least four good observations of P phases per event are obtained, which ensure more robust hypocentre solutions (in comparison, for instance, to events located with S-wave observations only).

## Survey of the impact on population and infrastructure

To quantify the impact of the lava flows and better guide the humanitarian response, a field survey was designed and organized with the Civil Protection and the National Institute of Statistics of the North Kivu province in the immediate aftermath of the eruption. Nineteen representatives of these two institutions used an online survey platform and their knowledge of the field to assess the impact on both the main infrastructure and the population directly affected by the lava flows. The survey was organized in two different but complementary phases: the first phase (2 days; 1 and 2 June 2021) aimed at a global evaluation of the impact per village (damaged infrastructure and human impact), with the help of the local authorities; it constituted a baseline for the second phase (11 days; 3–13 June 2021), targeting the affected population for more detailed information collection. Communication and consultation with local authorities (that is, the village chiefs) were at the heart of the methodology and played a crucial role in the success of this survey.

It should be noted that the exact number of people who died has yet to be determined. Our survey, conducted in the emergency phase, was unable to differentiate between the dead and the missing (220 persons). So far, the official number of fatalities remains the one published by the United Nations Office for the Coordination of Humanitarian Affairs in its first situation report on 26 May 2022, stating 31 deaths, of which about one third were victims of road accidents during the evacuation.

In addition, our lava flow impact survey revealed that approximately 6,000 households were left homeless, of which it is important to distinguish between homeowners and renters: about 4,700 versus 1,200 households, respectively. Since July, the Provincial Division of Humanitarian Actions and National Solidarity (DIVAHS) of North Kivu has been in charge of the humanitarian response. Owing to the difficulties in taking over the crisis management, the DIVAHS carried out its own estimate of the number of homeless households, focusing only on homeowners. Close to the results of our survey, this assessment determined that about 4,200 households needed a new shelter.

## GNSS processing

Long-term data processing from the KivuGnet GNSS network is performed fully automatically using 24-hour long files sampled every 30 s. Double-difference observations are used for the calculation of precise daily coordinates in the Nubia-fixed reference frame using Bernese v.5.2 software[50]. The complete data processing strategy is described in ref. [25]. Owing to the large amplitude and short timescale of the co-eruptive displacements, specific kinematic processing of GNSS data was performed in addition to the 24-hour processing. The GIPSY-OASIS-II (processing engine is based on a Kalman filter approach and can output single-station solutions (precise-point-positioning) with centimetre-scale accuracy in a global reference frame at a high rate[51]. The GIPSY-OASIS-II software was used with 24-hour files sampled every 30 s to output position solutions at 5 min intervals. For the analysis we used ultra-rapid satellite orbits and clock information from the Jet Propulsion Laboratory (http://sideshow.jpl.nasa.gov/pub/JPL_GPS_Products/Ultra/), IGS antenna calibrations and the FES2004 ocean tidal loading[52].

## SAR and InSAR computation

The last pre-eruptive interferogram (Fig. 4) was computed with COSMO-SkyMed (CSK) Stripmap images acquired in the X-band (3.1 cm wavelength) one day before the eruption (21 May 2021 at 15:37 UTC) and less than an hour before the eruption (22 May 2021 at 15:37 UTC) along a descending orbit. The interferogram, which has a perpendicular baseline of around −376 m, was computed using MasTer[53,54]. Images were multilooked two times in range and azimuth. Similarly, a descending CSK interferogram spanning the period 30 May 2021 to 7 June 2021 was also computed for mapping purposes.

Co-eruptive S1 interferograms were computed with MasTer using interferometric wide-swath C-band images (5.5 cm wavelength) acquired on 19 May 2021, 25 May 2021 and 31 May 2021 along the ascending orbit 174 and 21 May 2021, 2 June 2021 and 14 June 2021 along the descending orbit 21. Because deformation sprawled over a large area, the MasTer toolbox stitched 17 bursts spanning two frames from images acquired along ascending orbits and 16 bursts from images acquired

along descending orbits. No multilooking was applied to keep the highest spatial resolution and hence image at the best high deformation gradient.

The topographic phase contribution was removed on the basis of the Shuttle Radar Topography Mission (SRTM) digital elevation model. Unwrapping of the S1 interferograms was performed with SNAPHU[55] using a recursive procedure. Each of the 15 iterations consisted of unwrapping the interferogram then applying a low-pass filter cutting wavelengths smaller than a threshold. The unwrapped filtered interferogram was rewrapped and subtracted from the initial interferogram while the cutting frequency increased by 90%. Wherever possible, we ensured that we got consistency between GNSS data projected along the LOS and the unwrapped phase.

In addition, ascending and descending interferograms spanning the eruption were processed with the Interferometric Synthetic Aperture Radar (InSAR) Scientific Computing Environment (ISCE[56]) using SAR images from the Japanese Space Agency's (JAXA's) L-band satellite ALOS-2 (23.8 cm wavelength). The ascending pair (track 179) was calculated using two stripmap (SM3) images spanning 30 July 2020 to 3 June 2021. It was multilooked four times in range and eight times in azimuth, and ionospheric corrections were applied using the range split-spectrum method implemented in ISCE before unwrapping[57].

We compared the deformation measured by InSAR and GNSS (projected into the InSAR LOS) to assess the accuracy of the ionospheric corrections. If the location of the GNSS station was not coherent in the interferogram, we manually chose a coherent location that had a similar displacement, based on the fringe rate. We found that an ionospheric phase estimation computed using a multilooking of 64 looks in range, 128 looks in azimuth, a minimum filter window size of 51 and a maximum filter window size of 151 resulted in consistent measurements between the ascending interferogram and the GNSS displacements. The descending pair was calculated using an SM3 (track 77) image acquired on 14 May 2021 and a ScanSAR (wide-swath mode, WD1) image acquired on 28 May 2021. The interferogram was processed using 4 looks in range and 14 looks in azimuth, and ionospheric corrections were computed using multilooking of 8 looks in range and 56 in azimuth, as well as a minimum filter window size of 51 and a maximum filter window size of 151. The InSAR and GNSS measurements were consistent, aside from a vertical offset of around 10 cm at stations KBT and RSY. A weighted power spectrum filter and cascading high-pass filter were applied before unwrapping the interferograms with an iterative, coherence-based algorithm. These algorithms were implemented as modules in NSBAS[58,59]. The unwrapped interferograms were masked on the basis of unwrapping iteration (10,000 for ascending and 14,000 for descending). Fringe discontinuities related to graben faulting were observed in both the ascending and descending interferograms, and the eastern portion of the ascending interferogram was shifted by $-2\pi$ to account for unwrapping errors, ensuring a far field with an average displacement of zero. Regions with remaining unwrapping errors were manually masked.

## Cartography

Lava flow and fissure mapping to assist the interpretation of the eruption and the related impacts started in the field directly after the eruption and using space- and air-based imagery as soon as they were available.

The mapping of the 2021 lava flows of Nyiragongo were successively performed and improved using structure-from-motion photogrammetry, and a video taken during a helicopter flight of the United Nations Organization Stabilization Mission in the DR Congo (MONUSCO), Sentinel-1 VH-polarized radar amplitude images (with ground sampling distance (GSD) of 10 m) acquired by the S1 satellites, post-eruptive coherence images (GSD = 4 m) derived from the CSK InSAR processing, Sentinel-2 multispectral images (GSD = 10 m), Planetscope[46] multispectral images (GSD = 3.7 m) and, finally, a very-high-resolution

(GSD = 6.5 cm) orthophotograph produced with photographs acquired during a dedicated helicopter flight on 4 June 2021, using structure-from-motion photogrammetry and the Agisoft Metashape (v.1.6.5) software.

The eruptive fissures were mapped in detail using the Planetscope[46] images and the very-high-resolution orthophotograph produced. The ground fissures were mapped from S1 and CSK co- and post-eruptive wrapped interferograms computed with MasTer[53,54].

## Crater shape from SAR analysis

SAR imagery from a variety of satellite-borne sensors (Extended Data Table 2) provide subdaily observations of the Nyiragongo summit and enable the tracking of the evolution of the crater collapse. Assuming that the crater has a nearly circular shape and constant rim elevation, the crater rim has an elliptical shape in the SAR image geometry. By mapping the semi-axes of the ellipse, and knowing the pixel size and incidence angle of the images, the area of the crater can be calculated from every single image. The depth of the crater is deduced by picking the slant-range location of the bottom of the crater, comparing it to the slant-range location of crater edge and assuming an internal conical shape with an angle of 30°. Depth values are given in reference to the 2002 remnant platform P2 at around 3,190 m.

## Ash and $SO_2$ detection

Space measurements of $SO_2$ and aerosols were processed in near real time during the crisis. The total mass of $SO_2$ from the TROPOspheric Monitoring Instrument (TROPOMI) hyperspectral sensor onboard the Sentinel-5 Precursor European Space Agency (ESA) satellite[60,61] was estimated in an automated way within the SACS monitoring system[62]. Daily overpass of TROPOMI over the Kivu region is at about 11:30 UTC. Here we assumed the height of the $SO_2$ plume was at 4 km. To retrieve the total $SO_2$ mass over the Virunga region, we used the same pixels selection as in ref. [19].

Data from the Spinning Enhanced Visible and InfraRed Imager (SEVIRI) broadband sensor, onboard the geostationary satellite MSG-4, was used to detect ash plumes with a temporal resolution of 15 min. Considering the eight thermal infrared channels of SEVIRI, the spectral significance of ash in the data was estimated for each SEVIRI pixel in the form of an ash index[63]. To obtain a proxy of the volcanic activity, the sum of the ash index was calculated for each 15-min scan, considering all pixels within a 15 km radius around Nyiragongo.

## Dyke modelling

We used the elastic boundary element method and inversion implemented in DEFVOLC to determine the dyke geometry and overpressure[64] from S1 and ALOS displacement data. Topography is meshed using the SRTM-1 digital elevation model[65] and bathymetry data[66]. The mesh is refined in the vicinity of the eruptive fissures and enlarged in the far field for computation ability. InSAR along LOS displacements are shifted to have a reference of zero displacement on average in the far field. Low or non-coherent areas including Lake Kivu are masked out. Because the deformation associated with the graben formation is mostly inelastic[20], as we used linear elastic modelling, we also masked the corresponding area in Goma and Gisenyi. Note that the graben fault slip is also expected to influence the displacement fields outside of the graben. Not including those faults in the modelling could be a reason for some of the observed residuals in the vicinity of the dyke. Data are subsampled at the locations of topographic mesh nodes. Spatially correlated noise in InSAR data is taken into account through the use of a full covariance matrix to weight the data in the cost function computation[64].

The deformation source is defined as a quadrangle linked to the surface[64] by six 'en echelon' fractures at eruptive fissure locations. The height of the connecting fractures and the characteristics of the bottom line of the quadrangle lead to seven inversion parameters (Extended

Data Table 1 and Extended Data Fig. 5). In addition, the pressure is linearly inverted[67]. The direct problem is solved using the mixed boundary element method[21,22]. The non-linear inverse problem resolution is performed with an implementation of the Neighborhood algorithm[64,68]. Posterior probability density functions and model uncertainties are calculated following the Bayesian inference framework[64,69] (Extended Data Fig. 6 and Extended Data Table 1).

## X-ray fluorescence analysis

Major element compositions of whole-rock samples were determined through X-ray fluorescence (XRF) using an ARL PERFORM-X 4200 (Rh X-ray tube) spectrometer at the University of Liège. Samples were powdered at KU Leuven, Belgium by first crushing the samples manually, after which a Pulverisette planetary micromill was used to obtain a grain size of less than 1 μm. The resulting powders were dried at 1,000 °C for 2 h to remove organics and determine the loss on ignition. The dried powder was mixed with lithium tetraborate at a ratio of 1:11 to produce glass discs for major element analysis. Major element calibration was performed using 66 international standards (magmatic whole rocks, minerals and soil samples). Accuracy was determined to be better than ±2% for all elements except $Na_2O$ (±8.1%) and MnO (±4.4%). Likewise, repeated analyses indicate the $1\sigma$ analytical precision to be within 5% for major elements, with the exception of $Na_2O$ (6.6%). An additional 7 g of unheated material was used to make pressed powder pellets on which select trace elements were measured (V, Cr, Co, Ni, Cu, Zn, Ga, Rb, Sr, Y, Zr, Nb, Ba, Ce and Th). Trace element data determined by XRF are primarily used for comparison with inductively coupled plasma mass spectrometry, with the exception of Nb and Zr, for which XRF data are preferred. The quality of the trace element acquisition was estimated by repeated measurement of 11 international standards (GSD-10, GSD-9, JF-1, JR-1, Nim-D, Nim-G, Nim-L, Nim-N, Nim-P, STM-1 and SY-2). Accuracy is estimated to be better than ±10% for all elements except for Cr (±19%), Co (±20%), Ba (±19%) and Ce (±24%). A $1\sigma$ analytical precision is better than ±3% for all elements.

## Electron microprobe analysis

Electron microprobe analysis was performed at Utrecht University using a JEOL JXA-8530F Hyperprobe. Operating conditions were 15 kV for all analysis, using a nominal beam current of around 15 nA and 10 nA. On-peak counting times of 20 s were used for all elements, with off-peak counting times of 10 s before and after analysis. Na and K were always measured first to avoid alkali loss during analysis. Secondary standardization was performed on Smithsonian Kakanui augite (NMNH 122142), San Carlos olivine (NMNH 111312-44), ilmenite (NMNH 96189) and microcline (NMNH 143966). Relative errors on major element concentrations are generally less than 5% for all concentrations above 0.1 wt%. Specifically, errors are 0.8% for $SiO_2$, 1.8% for $Al_2O_3$, 2.5% for MgO, 4% for $TiO_2$, 1.8% for FeO, 1.4% for CaO, 7.6% for $Na_2O$, 3.3% for $K_2O$ and 2.2% for MnO at these concentrations.

## Thermal analysis

During the 2021 eruptive crisis, the HOTSAT thermal monitoring system[70,71] provided the location and quantification of the thermal anomalies by computing the radiant heat flux from images of the MODIS, SEVIRI and VIIRS satellite sensors. Extended Data Fig. 7 shows the behaviour of the radiant heat flux from 1 April to 30 June 2021 over Nyiragongo volcano. All the satellite sensors are coherent and measure moderate values of the radiant heat flux around 1–1.5 GW, which are typical values normally measured on that lava lake[72]. The only exception is represented by higher values registered on 15 May 2021 both by SEVIRI (up to 2.57 GW) and VIIRS. It is worth noting that during the eruptive crisis most of the thermal activity was totally or partially covered by the dense volcanic plume so it is underestimated. After 26 May, almost no thermal anomalies were observed.

## Data availability

Data used in this study are available for download from https://doi.org/10.5281/zenodo.6539427. The SAR imagery that supports the findings of this study is available from the space agencies and satellite operators (ESA/Copernicus, DLR, JAXA, ASI, CNES/Airbus, CSA, CONAE, ICEYE, Capella, Planet) but restrictions apply to the availability of some of these data, which were used under license for the current study, and so are not systematically available publicly. Data are, however, available from the authors upon reasonable request and with permission of the relevant data provider. Sentinel data are made available by ESA, https://scihub.copernicus.eu/. SAR images of Nyiragongo were accessed through the International Charter 'Space and Major Disasters' activation no. 713 https://disasterscharter.org/web/guest/activations/-/article/volcano-in-congo-the-democratic-republic-of-the-activation-713-, with exception of Capella imagery, which may be accessed through Capella's Open Data programme https://www.capellaspace.com/community/capella-open-data/. Source data are provided with this paper.

## Code availability

DEFVOLC software is available at http://www.opgc.fr/defvolc/Vue/.

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

**Acknowledgements** Sentinel-1 SAR images and Sentinel-5P TROPOMI data are provided by ESA. ALOS-2 SAR images are provided by JAXA under the terms and conditions of the Second Earth Observation Research Announcement (PI No. ER2A2N086). We acknowledge S. Sobue and Y. Aoki for assisting in the quick response of JAXA. CSK images were obtained from ASI through the GEO GSNL Supersite initiative. We thank M. Poland, S. Ebmeier and M. Bonano for having helped unlock the delivery of CSK images to the Virunga Supersite, and C. Tinel and C. Proy (French Centre National d'Etude Spatiales (CNES)) for facilitating coordination between the scientific response and space agencies participating in the United Nation (UN) 'Space and Major Disasters' International Charter. TerraSAR-X SAR images were obtained from DLR through the International Charter 'Space and Major Disasters' (© DLR e.V. 2021, Distribution Airbus DS Geo GmbH). Pléiades images were provided by the French CNES in the frame of DINAMIS project no. 2021 123 and ForM@Ter's CIEST2 activation. PlanetScope images were provided through the Planets Education and Research Standard plan (ID 81527/PI:R.G. and ID 581018/PI:BS). We thank S. Ebmeier for facilitating access to ICEYE satellite imagery. We thank the Capella company for rapid tasking. L. Clarisse and N. Clerbaux helped obtain SEVIRI data provided by EUMETSAT. GNSS and seismic data were provided by the KivuSnet and KivuGnet monitoring networks maintained by European Center for Geodynamics and Seismology/ Musée National d'Histoire Naturelle (MNHN Luxembourg), the Royal Museum for Central Africa, the Goma Volcano Observatory, the Centre de Recherche en Sciences Naturelles de Lwiro, the Université Officielle de Bukavu, the Rwanda Petroleum and Mining Board (RMB) and the University of Bujumbura (Burundi). The Congolese Institute for Nature Preservation, MONUSCO and GVO staff provide support to ensure the security of these stations. The permanent ground-based monitoring infrastructures and the contributions from B.S., C.M. and J.B. benefited from several past and ongoing research projects funded by the STEREOIII Programme of the Belgian Science Policy Office, the Fonds National de la Recherche of Luxembourg and the Belgian Directorate General for Development Cooperation and humanitarian aid, a.o. RESIST (SR/00/305 and INTER/STEREOIII/13/05), VeRSUS (SR/00/382) and HARISSA. RMB assisted the deployment of the temporary seismic stations. Airborne campaigns were made possible with the support of MONUSCO. J. L. Froger and Y. Fukushima provided the unwrapping algorithm used for S1 interferograms. Y. Morishita, P. Lundgren and F. Delgado helped improve the processing and ionospheric corrections of the ALOS-2 interferograms. Deformation modelling benefited from the infrastructure and assistance of the Mésocentre from the University of Clermont Auvergne, from funds from French Government Laboratory of Excellence initiative no. ANR-10-LABX-0006. This is Laboratory of Excellence ClerVolc contribution no. 548. This work is a contribution to the EUROVOLC project, under the EU Horizon 2020 and Innovation Action, grant no. 731070. S.P. was supported by a FRS-FNRS postdoctoral fellowship at Université libre de Bruxelles . C.W. acknowledges funding by the National Science Foundation (grant no. EAR 1923943).

**Author contributions** D.S., B.S., A.O., J.B., N.d.O., C.M. and F.K. wrote the paper. J.B., A.O., C.C. and J.S. performed the seismic and infrasound analysis. D.S., T.S., N.d.O., D.D. and H.L.M. performed the InSAR analysis. H.G. and N.d.O. performed the GNSS analysis. D.S., T.S., C.W. and V.C. performed the geodetic modelling. D.S., T.S., R.G., N.d.O., D.D. and J.B. performed the SAR image analysis for the geomorphology and cartography. B.S., L.D. and S.P. performed the optical and multispectral image analysis for the geomorphology and cartography. N.T., H.B. and P.d.B. performed the SO$_2$ and ash analysis. G.G. and B.S. performed the thermal analysis. O.N., E.K.K., S.M. and B.S. performed the rock sampling and petrological/chemical analysis. C.M., O.C., M.K., C.K.R., J.K.M. and I.K.N. performed the field surveys and impact assessment. F.D. performed the Lake Kivu water monitoring. F.D. and M.S. contributed to the assessment of the risk of a limnic eruption. M.Y., P.A., C.K.M. and A.S.M. assisted with the crisis management and field work coordination. All authors interpreted and discussed the results, and edited, reviewed and approved the manuscript.

**Competing interests** The authors declare no competing interests.

**Additional information**
**Correspondence and requests for materials** should be addressed to D. Smittarello.

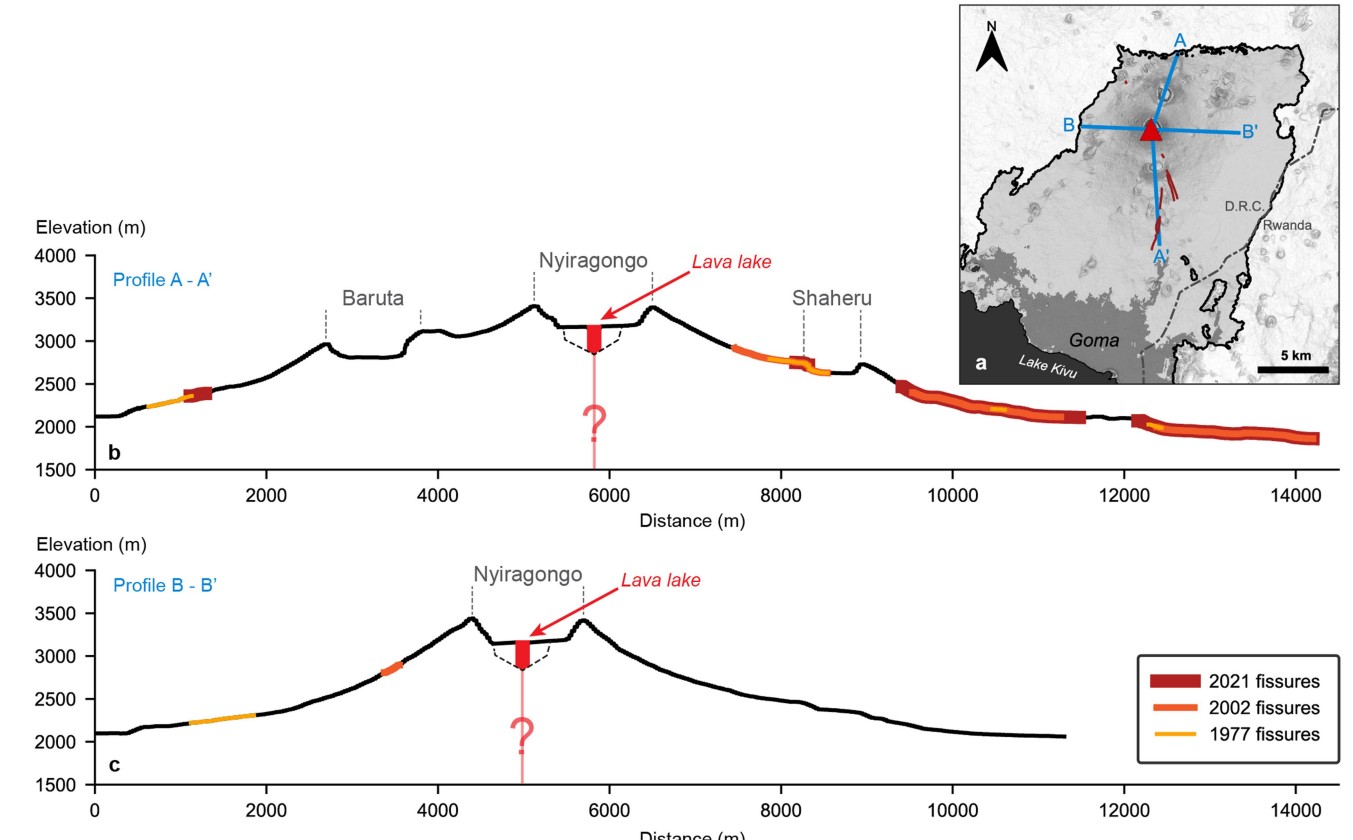

**Extended Data Fig. 1 | Cross sections of the Nyiragongo edifice.** (a) Slope map derived from SRTM-1 digital elevation model showing the location of the 2021 eruptive fissures (brown lines). (b) North-South (A-A') and (c) East-West (B-B') profiles across Nyiragongo edifice locations are shown by blue lines on map (a). Projection of the 2021, 2002 and 1977 eruptive fissure location on both profiles are represented with brown, orange and yellow lines respectively.

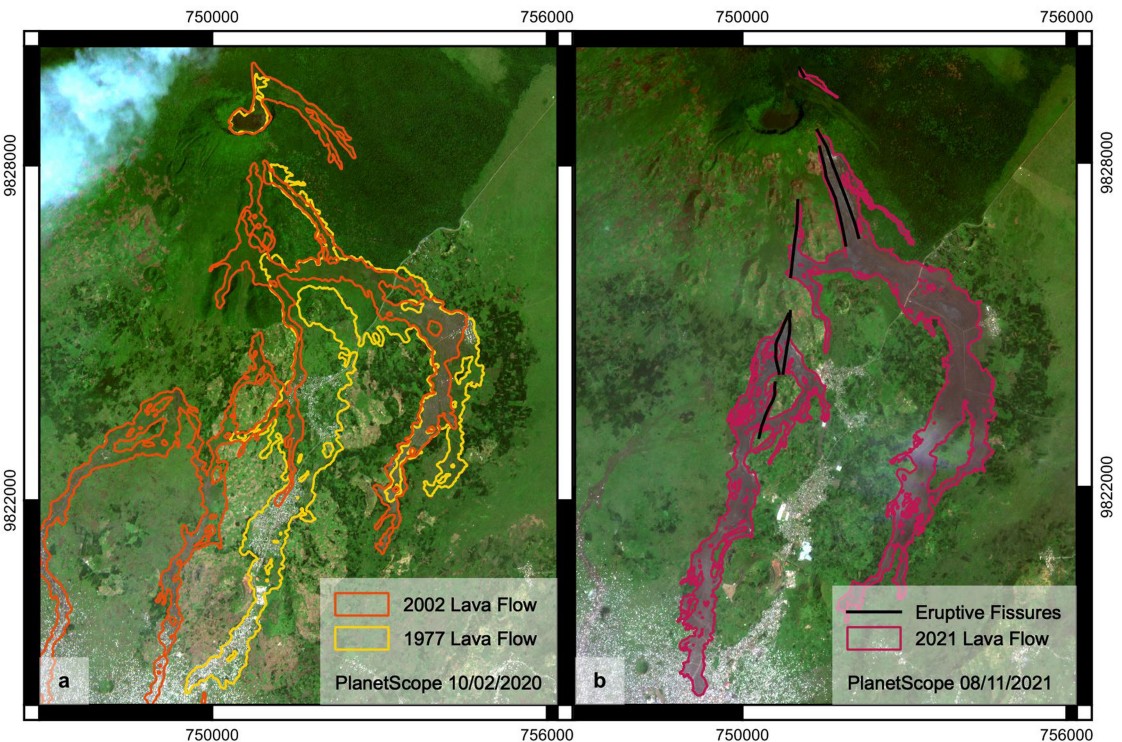

**Extended Data Fig. 2 | Nyiragongo's lava flows.** Paths of the lava flows of the two previous eruptions (1977 in yellow and 2002 in orange on map (a)) in comparison with the path of the lava flow of the 2021 eruption (in red on map (b)) are overlaid on PlanetScope images from 10/02/2020 (a) and 08/11/2021 (b), respectively.

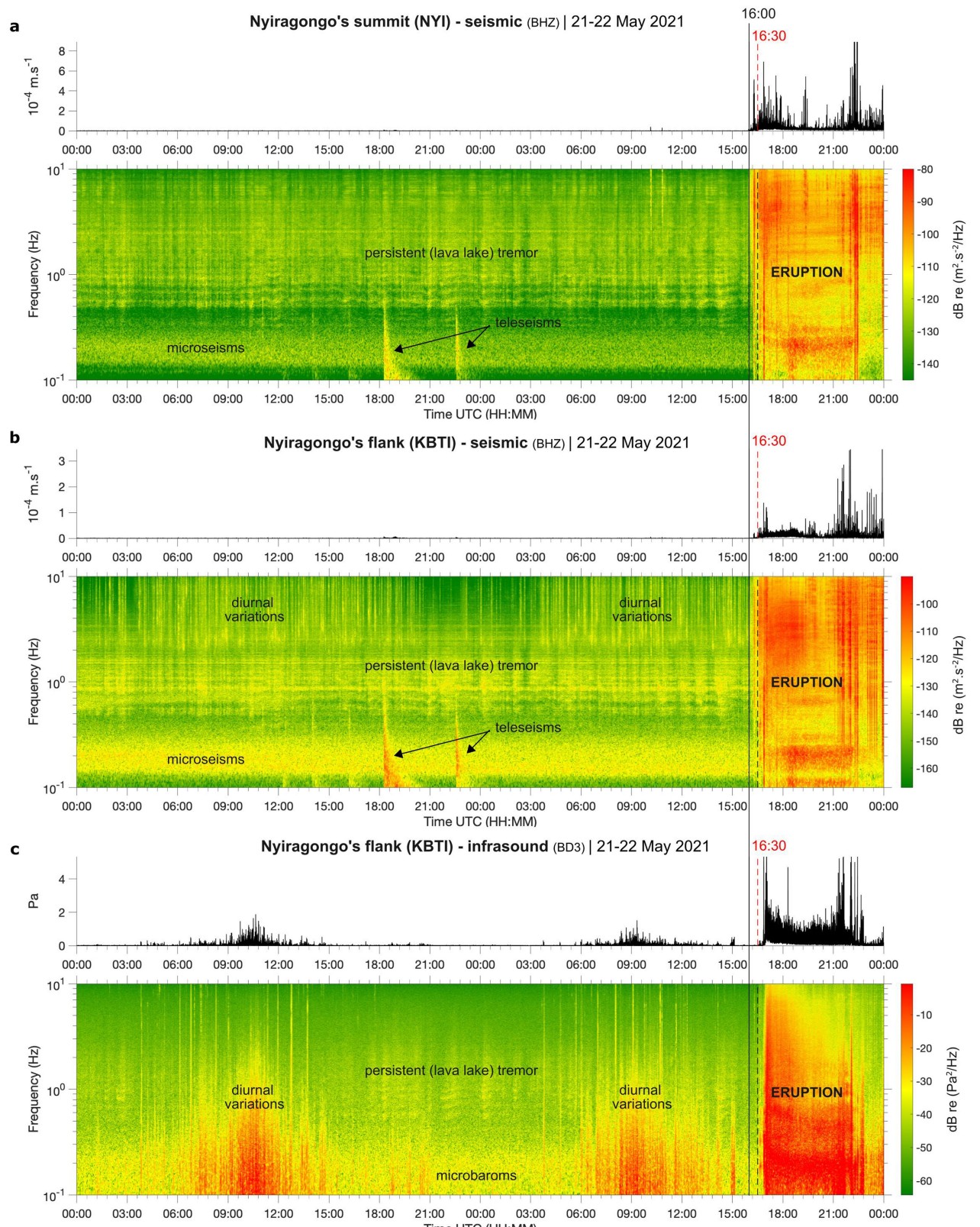

**Extended Data Fig. 3 | Seismic and infrasound records.** Seismic and infrasound traces on 21-22 May 2021 filtered within the frequency band [0.1-10] Hz (1-sec moving median of absolute envelope values corrected from the instrumental gain) and corresponding spectrograms (PSD - Power Spectral Density using 5-min time window with 50% overlapping) obtained from a) the seismic station NYI at Nyiragongo's summit (i.e., located on the main crater's rim), b) the seismic and c) acoustic station KBTI deployed on Nyiragongo's eastern flank about 6 km away from the summit. The absolute envelope values are truncated below the 99.99th percentile and the PSD values (in dB) are encompassed between the 5th and 99.9th percentiles. Indications onto the spectrograms are added to help interpreting the main seismic (a and b) and infrasound (c) patterns recorded at the stations (both are the closest to the eruptive fissures among the stations from the KivuSNet network). The broadband, persistent seismo/acoustic tremor from the lava lake activity above 0.4 Hz (e.g., Barrière et al., 2019) and the diurnal variations mainly due to the human activity are clearly visible. The timing of the first (high-frequency > 2Hz) seismic signals detected at the summit around 16:00 UTC and the first evident volcanic infrasound signals detected on the flank after 16:30 UTC related to the eruption are highlighted by solid and dashed lines, respectively (see the main text for more details).

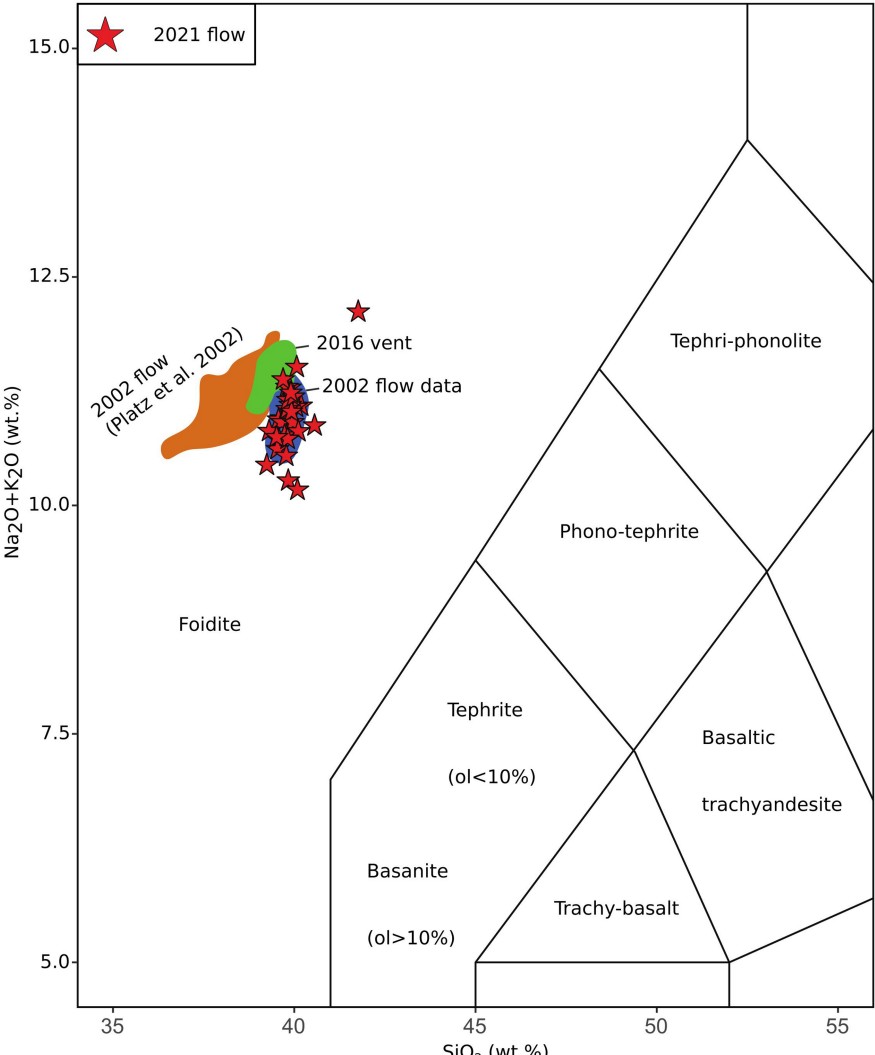

**Extended Data Fig. 4 | Total Alkali Silica diagram.** Lavas from Nyiragongo eruption 2002 are represented in blue (this study) and orange[73]. Lavas from the intra-crater vent appeared in 2016 are in green. Lava samples from Nyiragongo 2021 eruption are represented by red stars.

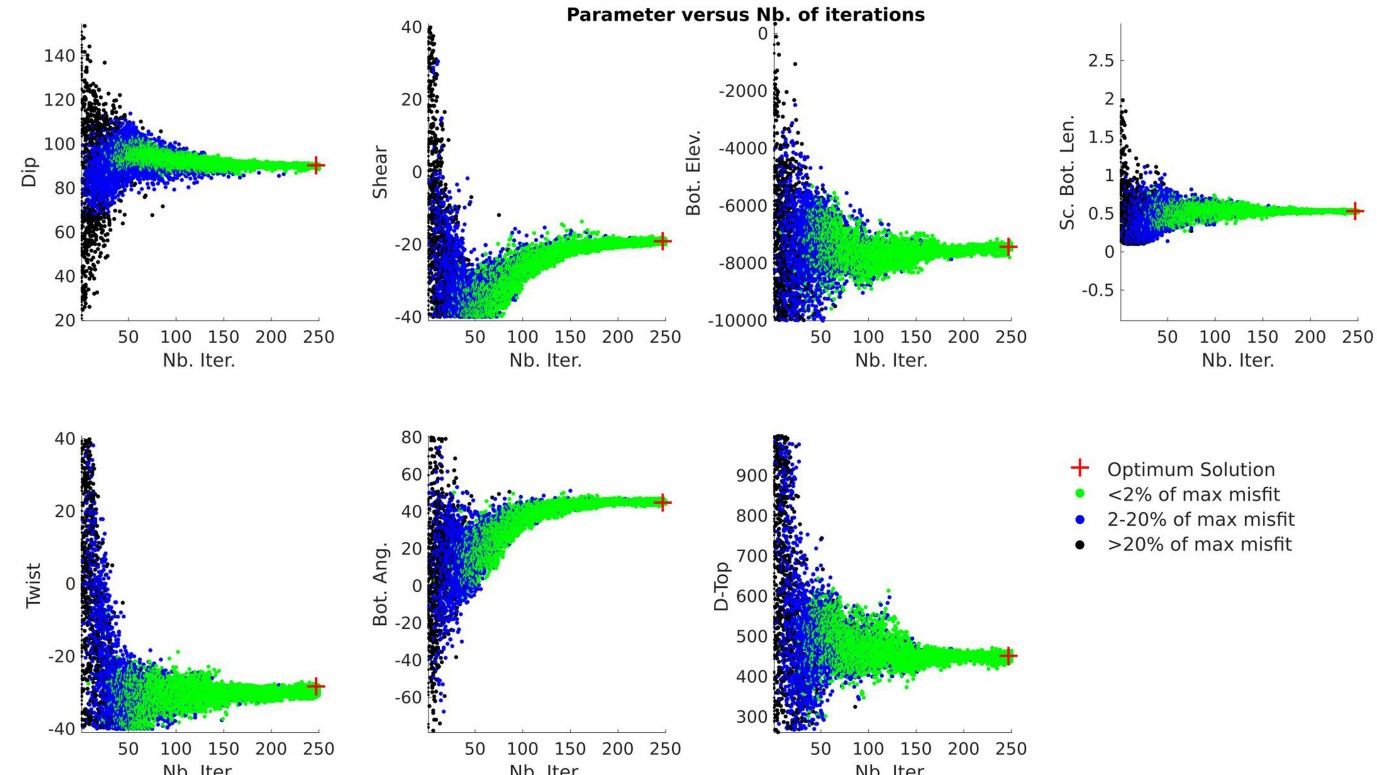

**Extended Data Fig. 5 | DEFVOLC inverted parameters.** Evolution of the 7 model parameters inverted with DEFVOLC as a function of the iteration number. Red cross represents the best model. Black, blue and green dots represent models whose misfit is >20%, ranges in 2-20% or is <2% of the misfit corresponding to a null model, respectively.

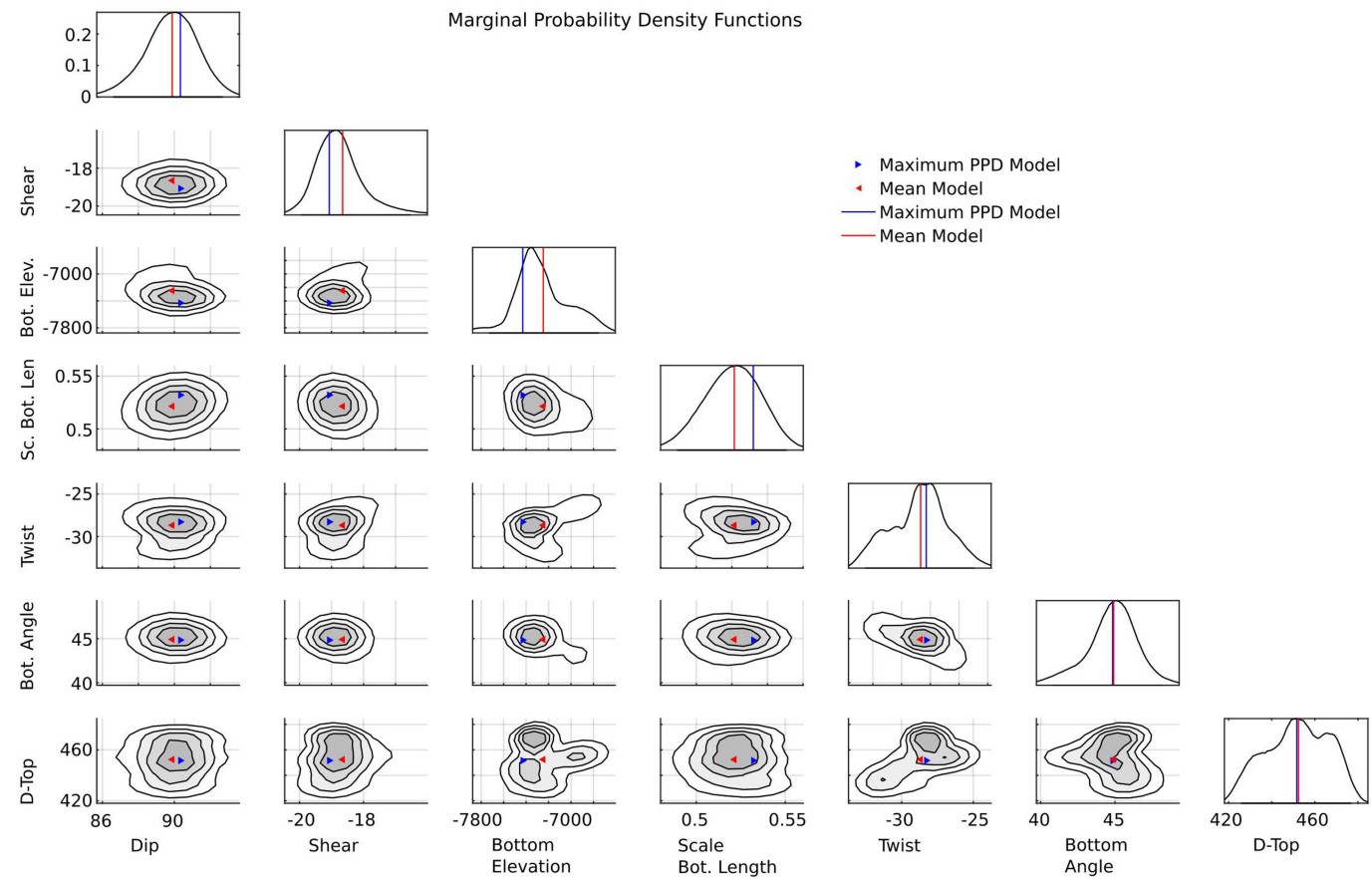

**Extended Data Fig. 6 | Inversion results.** One-dimensional (diagonals) and two-dimensional (off-diagonals) marginals posterior probability density functions.

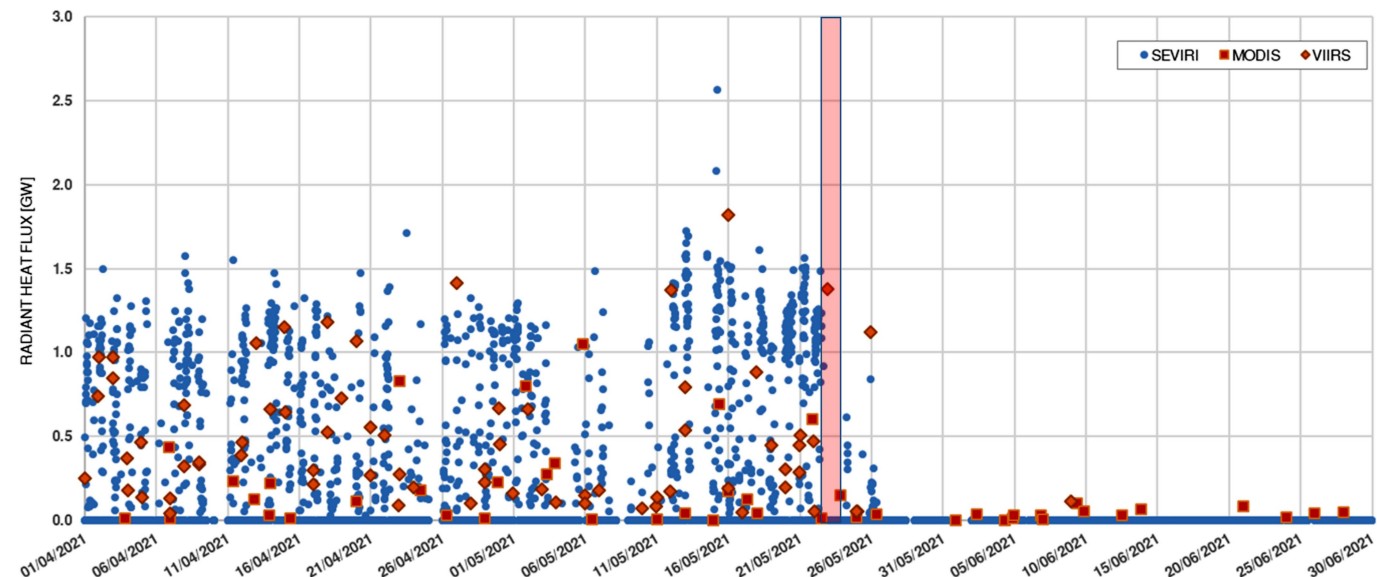

**Extended Data Fig. 7 | Thermal anomaly.** Radiant Heat Flux derived from SEVIRI (blue dots), MODIS (red squares) and VIIRS (orange rhombus) over Nyiragongo area during 1 April to 30 June 2021. Absolute values are challenging to interpret, as the thermal monitoring of Nyiragongo is complicated by the presence of a thick volcanic gas plume and frequent cloud cover. The highest values are generally associated with large lava lake overflows and/or less cloudy conditions.

## Extended Data Table 1 | Inversion parameters

| Inputs | Values | Description |
|---|---|---|
| E | 5 GPa | Young Modulus |
| ν | 0.25 | Poisson ratio |
| D | 850 m | InSAR correlation distance |
| σ | $5.10^{-4}$ $m^2$ | InSAR data variance |
| NS1 | 500 | Sample size for first iteration |
| NS2 | 50 | Sample size for others iteration |
| NR | 50 | Number of cells to resample |
| NIT | 250 | Maximum number of iterations |
| NLST | 50 | Number of last misfit values that are used to evaluate the standard deviation |
| STDDEV | 0.3 | Standard deviation threshold for the inversion termination |
| NPRL | 50 | Number of forward computations that run in parallel |

| Parameters | Explored Range | Confidence Interval | Best value | Mean Value ± Standard Deviation |
|---|---|---|---|---|
| Dip (°) | [0 : 180] | [86.6 : 92.7] | 90 | 89.9±1.5 |
| Shear (°) | [-40 : 40] | [-19.9: -16.5] | -19 | -18.7±0.8 |
| Bottom Elevation (m) | [-10,000 : 1000] | [-7,729 : -6,752] | -7,430 | -7,247±234 |
| Scaled Bottom Length | [0.1 : 2] | [0.49 : 0.55] | 0.53 | 0.52±0.02 |
| Twist (°) | [-40 : 40] | [-32.5 : -24.9] | -28 | -28.7±1.9 |
| Bottom Angle (°) | [-80 : 80] | [40.8 : 48.2] | 44 | 44.9±1.7 |
| Dtop (m) | [100 : 1,000] | [425 : 477] | 451 | 452±14 |
| Pressure (MPa) | Linearly Inverted | | 0.8 | |

Modeling input values and explored range and best values of the inverted parameters.

**Extended Data Table 2 | Crater dimensions resulting from SAR imagery analysis**

| Satellite | Date YYYY-MM-DDTHH:mm:ss | Incidence (°) | Crater Radius (m) | Crater Depth (m) |
|---|---|---|---|---|
| ALOS2 | 2020-03-06T09:41:00 | 41.0 | / | / |
| ALOS2 | 2020-07-30T22:20:29 | 40.2 | / | / |
| CSK | 2021-03-19T15:37:30 | 26.5 | / | / |
| CSK | 2021-05-16T04:03:41 | 35.0 | / | / |
| Sentinel 1 | 2021-05-19T16:21:00 | 38.0 | / | / |
| CSK | 2021-05-21T15:37:00 | 26.5 | / | / |
| CSK | 2021-05-22T15:37:00 | 26.5 | / | / |
| CSK | 2021-05-23T04:03:41 | 35.0 | 228 | 129 |
| Capella | 2021-05-23T19:28:48 | 30.2 | 293 | 281 |
| RCM | 2021-05-24T03:46:00 | 40.8 | 358 | 350 |
| TSX | 2021-05-24T16:13:36 | 26.0 | 355 | 327 |
| RCM | 2021-05-24T16:21:22 | 41.5 | 347 | 379 |
| Sentinel 1 | 2021-05-25T16:21:00 | 38.0 | 375 | 409 |
| RCM | 2021-05-25T16:29:19 | 53.5 | 407 | 449 |
| Capella | 2021-05-26T04:39:45 | 34.4 | 513 | 430 |
| ICEYE . | 2021-05-26T20:25:18 | 24.0 | 519 | 407 |
| SAOCOM | 2021-05-27T04:21:26 | 30.0 | 507 | / |
| ALOS2 | 2021-05-28T09:41:00 | 41.0 | 538 | 520 |
| CSK | 2021-05-30T15:37:30 | 26.5 | 481 | 567 |
| Sentinel 1 | 2021-05-31T16:20:00 | 38.0 | 500 | 511 |
| CSK | 2021-06-01T04:03:41 | 35.0 | 544 | 496 |
| Sentinel 1 | 2021-06-02T03:44:00 | 40.0 | 500 | 565 |
| ALOS2 | 2021-06-03T22:20:29 | 40.2 | 519 | 500 |
| TSX | 2021-06-04T16:13:13 | 26.0 | 498 | 476 |
| Sentinel 1 | 2021-06-06T16:21:00 | 38.0 | 542 | 485 |
| CSK | 2021-06-07T15:37:30 | 26.5 | 500 | 567 |
| Sentinel 1 | 2021-06-08T03:44:00 | 40.0 | 542 | 565 |
| CSK | 2021-06-08T04:03:41 | 35.0 | 551 | 488 |
| CSK | 2021-06-09T04:03:41 | 35.0 | 551 | 488 |
| TSX | 2021-06-11T03:57:11 | 21.5 | 550 | 541 |
| Sentinel 1 | 2021-06-12T16:21:00 | 38.0 | 542 | 485 |
| Sentinel 1 | 2021-06-14T03:44:00 | 40.0 | 542 | 522 |
| TSX | 2021-06-15T16:13:41 | 26.0 | 544 | 476 |
| TSX | 2021-06-22T03:57:11 | 21.5 | 550 | 522 |