## [Peer Review File · Nature]

Manuscript Title: Precursor-free eruption triggered by edifice rupture at Nyiragongo volcano

Reviewer Comments & Author Rebuttals

Reviewer Reports on the Initial Version:

Referee #1:

The May 2021 eruption of Nyiragongo resulted in 220 fatalities and left 6,000 households homeless. Given the large population at risk to fast moving lava flows and the possibility of a liminic eruption of Lake Kivu, things could have been far worse. This paper shows clearly that, unlike previous eruptions of Nyiragongo, the 2021 eruption had no obvious intermediate term precursors, either seismic, deformation (GNSS or InSAR), or gas emission. Indeed, eruptions started < 40 minutes after first seismic signals were recorded. Given that monitoring and contingency planning were developed based on previous eruptions, the only warning was visible lava flows moving toward the population centers.

The authors, quite reasonably, attribute the lack of premonitory signals to the open vent character of Nyiragongo, with an active lava lake at the summit of the volcano. As magma entered the shallow system the lava level increased, in 2020 starting to inundate the remnant platform of the 2002 crater collapse. This pressurization would not be expected to give rise to wide spread deformation (although this deserves future modeling) and stress changes were either insufficient to trigger earthquakes, or were concentrated at such shallow depths that the response was aseismic. However, the stress change was apparently sufficient to initiate fracturing and intrusion of a dike into the upper flank of the volcano. (Another possibility would be time dependent weakening due to sustained stresses above the long term background). The dike then propagated south toward Goma — as convincingly demonstrated by the evolving seismic swarm, GNSS displacements and InSAR. Elastic modeling suggests that the dike was within 500 meters of the surface through the populated areas of Goma. Only the highly degassed nature of the magma kept it from erupting into the city.

The paper is significant, convincing, and well presented. It is appropriate for publication in Nature. The summary point is very well taken: “Magma storage close to the surface in open systems means that eruptions may occur with only very short-term precursory activity, thus raising major challenges for their monitoring that need to be adequately addressed.”

I have only one significant suggestion: It would be very useful to have a N-S cross section of the upper part of the volcano showing the lava lake, the maximum lake level and the position of the high elevation fissures. (Something like a zoom in on Figure 3b). This would help to clarify how the lava lake rim was breached and possible dike paths at this elevation. This could go into the supplement, I leave that to the authors.

A few other points:

- 1) It seems that graben faults were not included in the modeling. While the authors excluded the graben area from the analysis, the fault slip will influence the displacement fields even outside the graben. I doubt this would make a major change to the interpretation, but it should be mentioned.
- 2) Line 164-5: "Crater collapse resulting from the reservoir drainage may in addition act as a piston, increasing magma flux and favouring the dike progression." I disagree. This happens when there is a solid roof (the piston) that collapses onto a magma reservoir, transferring weight onto the reservoir. It doesn't apply here with the open vent to the surface. Indeed as magma was withdrawn into the dike, the magma-static pressure must have decreased.
- 3) Figure 4c would be more useful to show at zoomed in scale focused on the pre-eruptive period. The figure makes clear that there were not large displacements, but it doesn't rule out small scale deformation.
- 4) Extended Figure 2: Are these fields supposed to be labeled?

Referee #2:

The authors present a summary of the 2021 eruption of Nyiragongo volcano. They analyze seismic, geodetic, geochemical data as well as mapping the impacts of the eruption and discuss future risks for nearby communities. This report provides a unique and comprehensive summary of these eruptive events. The manuscript is thorough by analyzing many different types of data. It is well written and the data appear to be analyzed with appropriate methods to support the conclusions. References are appropriate, but in some cases further details could be provided to show how they relate directly to Nyiragongo. I would support publication of this manuscript with very few minor changes, but there needs to be some discussion of uncertainties in the modeling results, and I do suggest expanding the discussion of global implications of this eruption for other volcanoes and advances in volcano science for Nature's broad audience.

The study of this eruption at Nyiragongo will help further the understanding of other global eruptions that lack precursory geophysical signals. For the broad audience of Nature, I think it would be beneficial for the authors to expand the discussion of the global analogues and implications, but I know word limits are tight. Expansion of the paragraph starting on page 156 would make those global implications much clearer and substantially improve the potential impacts of this paper. What lessons learned from other eruptions cited could apply to Nyiragongo? What lessons learned from Nyiragongo could be applied to other volcanoes? Some specific examples: the authors cite low buoyancy contrast references, but do the Nyiragongo magmas fall in this category? Does the reference to tectonic stresses sucking magma downrift apply to Nyiragongo ... is there evidence for tectonic spreading to support or decline this explanation? Although the 1977 and 2002 eruptions had tectonic triggers, the existence of recurring eruptions could enable forecasting abilities even in the absence of short-term geophysical precursors. For example, at Kilauea Montgomery-Brown and Miklius (Geology, 2020) show tensile failures with regular recurrence intervals to be a likely contributor of repetitive dike intrusions and raise the possibility of longer-term forecasting of future intrusions at Nyiragongo.

Uncertainties should be discussed in the geodetic data in particular because they are used to produce a model, and how those uncertainties propagate to the resulting model parameters.

Few small comments:

I appreciate the authors providing the geoid and UTM Zone, they are often left out of papers. While UTM zones are easy to look up, its one less step for the next researcher who wants to use the results from this paper.

It would be better to replace “complexified” with “complicated” in all cases.

Line 4 – replace “imply” with “appeal to”

Line 76 – Replace “Instead” with “In contrast”

Line 146 – remove “thus”

Line 151 - ” ... yet it would allow for some risk reducing actions”

Line 173 – Rephrase ... “The impacts of a magma intrusion within Lake Kivu is especially understudied.”

Fig 1 & Fig 4– the text on the map figure legends are very small and blurry on the screen and in print.

Line 581 – “subsamped at the locations of topographic nodes”

*****END*****

Author Rebuttals to Initial Comments:

Referee #1:

The May 2021 eruption of Nyiragongo resulted in 220 fatalities and left 6,000 households homeless. Given the large population at risk to fast moving lava flows and the possibility of a liminic eruption of Lake Kivu, things could have been far worse. This paper shows clearly that, unlike previous eruptions of Nyiragongo, the 2021 eruption had no obvious intermediate term precursors, either seismic, deformation (GNSS or InSAR), or gas emission. Indeed, eruptions started < 40 minutes after first seismic signals were recorded. Given that monitoring and contingency planning were developed based on previous eruptions, the only warning was visible lava flows moving toward the population centers.

The authors, quite reasonably, attribute the lack of premonitory signals to the open vent character of Nyiragongo, with an active lava lake at the summit of the volcano. As magma entered the shallow system the lava level increased, in 2020 starting to inundate the remnant platform of the 2002 crater collapse. This pressurization would not be expected to give rise to wide spread deformation (although this deserves future modeling) and stress changes were either insufficient to trigger earthquakes, or were concentrated at such shallow depths that the response was aseismic. However, the stress change was apparently sufficient to initiate fracturing and intrusion of a dike into the upper flank of the volcano. (Another possibility would be time dependent weakening due to sustained stresses above the long-term background).

We added a subsentence mentioning this alternative possibility lines 144-145. “we infer that the 2021 eruption was the consequence of an edifice rupture which could have resulted either from stress reaching tensile strength or from time-dependent weakening due to sustained stresses and elevated temperatures.”.

The dike then propagated south toward Goma — as convincingly demonstrated by the evolving seismic swarm, GNSS displacements and InSAR. Elastic modeling suggests that the dike was within 500 meters of the surface through the populated areas of Goma. Only the highly degassed nature of the magma kept it from erupting into the city. The paper is significant, convincing, and well presented. It is appropriate for publication in Nature. The summary point is very well taken: “Magma storage close to the surface in open systems means that eruptions may occur with only very short-term precursory activity, thus raising major challenges for their monitoring that need to be adequately addressed.”

I have only one significant suggestion: It would be very useful to have a N-S cross section of the upper part of the volcano showing the lava lake, the maximum lake level and the position of the high elevation fissures. (Something like a zoom in on Figure 3b). This would help to clarify how the lava lake rim was breached and possible dike paths at this elevation. This could go into the supplement, I leave that to the authors.

We thank the referee for this suggestion. We added such a N-S cross section as Extended Data Figure 1.

A few other points:

1) It seems that graben faults were not included in the modeling. While the authors excluded the graben area from the analysis, the fault slip will influence the displacement fields even outside the graben. I doubt this would make a major change to the interpretation, but it should be mentioned.

We agree with referee #1 that the fault slip may influence the displacement fields even outside the graben, mainly in the near field. Our model was mainly fit on the far field data, which makes up most of the subsampled data space. In addition to the inelastic deformation, this fault slip could explain some parts of the residuals where the model deviates from the data close to the dyke. We clarify in the Methods Dike Modeling paragraph that we do not include the graben fault in the modeling and now mention the graben fault slip influence as a possible cause for the observed residuals.

2) Line 164-5: "Crater collapse resulting from the reservoir drainage may in addition act as a piston, increasing magma flux and favouring the dike progression." I disagree. This happens when there is a solid roof (the piston) that collapses onto a magma reservoir, transferring weight onto the reservoir. It doesn't apply here with the open vent to the surface. Indeed as magma was withdrawn into the dike, the magma-static pressure must have decreased.

We would not be as categorical as referee #1 in stating that the piston concept does not apply here only because Nyiragongo is an open vent system. Specifically, sudden collapse of solidified material from the inner flank of the crater into the crater as magma is withdrawn from the system may induce a similar "piston" effect. But we believe that this point would merit further study and modelling and there is no room here for such a discussion. That's why we agree to remove this controversial sentence and associated references.

3) Figure 4c would be more useful to show at zoomed in scale focused on the pre-eruptive period. The figure makes clear that there were not large displacements, but it doesn't rule out small scale deformation.

We agree with this comment and propose a new version of the Figure 4 including zoom in showing no obvious short-term changes in deformation, seismicity nor gas time series.

4) Extended Figure 2: Are these fields supposed to be labeled?

Yes, we thank Referee 1 for pointing this out, we added labels to the fields in this figure which is now Extended Data Figure 4.

Referee #2:

The authors present a summary of the 2021 eruption of Nyiragongo volcano. They analyze seismic, geodetic, geochemical data as well as mapping the impacts of the eruption and discuss future risks

for nearby communities. This report provides a unique and comprehensive summary of these eruptive events. The manuscript is thorough by analyzing many different types of data. It is well written and the data appear to be analyzed with appropriate methods to support the conclusions. References are appropriate, but in some cases further details could be provided to show how they relate directly to Nyiragongo. I would support publication of this manuscript with very few minor changes, but there needs to be some discussion of uncertainties in the modeling results, and I do suggest expanding the discussion of global implications of this eruption for other volcanoes and advances in volcano science for Nature's broad audience.

We thank reviewer #2 for these constructive suggestions. We detail in the following how we added the discussion about uncertainties and briefly expanded the discussion including comparison with other volcanoes, while still complying with length constraints.

The study of this eruption at Nyiragongo will help further the understanding of other global eruptions that lack precursory geophysical signals. For the broad audience of Nature, I think it would be beneficial for the authors to expand the discussion of the global analogues and implications, but I know word limits are tight. Expansion of the paragraph starting on page 156 would make those global implications much clearer and substantially improve the potential impacts of this paper. What lessons learned from other eruptions cited could apply to Nyiragongo? What lessons learned from Nyiragongo could be applied to other volcanoes? Some specific examples: the authors cite low buoyancy contrast references, but do the Nyiragongo magmas fall in this category? Does the reference to tectonic stresses sucking magma downrift apply to Nyriagongo ... is there evidence for tectonic spreading to support or decline this explanation?

We appreciate the suggestion. We expanded the paragraph that now starts at line 153 to explicit the references to analogous dike propagation at others volcanoes and how the lessons learned apply to the case of Nyiragongo. We hope that it makes the global implications clearer while respecting the length constraints.

Although the 1977 and 2002 eruptions had tectonic triggers, the existence of recurring eruptions could enable forecasting abilities even in the absence of short-term geophysical precursors. For example, at Kilauea Montgomery-Brown and Miklius (Geology, 2020) show tensile failures with regular recurrence intervals to be a likely contributor of repetitive dike intrusions and raise the possibility of longer-term forecasting of future intrusions at Nyiragongo.

We thank the reviewer for this suggestion and agree that such a study at very active volcanoes like Kilauea or Piton de la Fournaise could help for longer-term forecasting but at Nyiragongo, the existence of only 3 flank eruption is known. Moreover, limitations in knowledge of the geological background prevent meaningful statistics. For this reason, we stated the following sentence already in the original version of the manuscript: "A probabilistic quantification of the associated risks is required, but remains highly challenging, in view of the limited knowledge on the eruption history of Nyiragongo.

Uncertainties should be discussed in the geodetic data in particular because they are used to

produce a model, and how those uncertainties propagate to the resulting model parameters. We are thankful for referee #2 pointing this out as we believe that the additional information about the uncertainties and how to account for these in the modeling will really strengthen our manuscript.

We now provide a concise description of these aspects in the dike modeling paragraph in the Methods section, i.e. how we account for the uncertainties through the cost function computation, where the covariance matrix is a full matrix for InSAR data. We also add some information on the two steps of the inversion (search and appraisal) and add the resulting Posterior Probability Density functions in a new Extended Data Figure 6. In order to respect the length constraints, we keep this description very concise and refer to publications containing all the details. We also added the Extended Data Table 1 containing the input values used in our inversion as well as explored and resulting confidence intervals. Finally, the zenodo archive contains all inputs, outputs and log files of the inversion.

Few small comments:

I appreciate the authors providing the geoid and UTM Zone, they are often left out of papers. While UTM zones are easy to look up, it's one less step for the next researcher who wants to use the results from this paper.

Thank you.

It would be better to replace "complexified" with "complicated" in all cases.

DONE

Line 4 – replace "imply" with "appeal to"

DONE

Line 76 – Replace "Instead" with "In contrast"

DONE

Line 146 – remove "thus"

DONE

Line 151 - " ... yet it would allow for some risk reducing actions"

DONE

Line 173 – Rephrase ... "The impacts of a magma intrusion within Lake Kivu is especially understudied."

DONE

Fig 1 & Fig 4– the text on the map figure legends are very small and blurry on the screen and in print. We increased the text size in both figures and provide separated editable figures (pdf) at full resolution. This should solve the blurry effect.

Line 581 – "subsamped at the locations of topographic nodes"

DONE

*****END*****

Reviewer Reports on the First Revision:

Referee #1:

The authors have done a very nice job of responding to the comments/suggestions of both reviewers. I believe the paper is now ready for publication.

Referee #2:

The authors appear to have addressed the reviewers comments. I have no further comments, and support its publication.

There is one last incidence of "complexified" at line 780 that should be changed to "complicated"

*****END*****